# Temperature- and Frequency-Dependent Ferroelectric Characteristics of Metal-Ferroelectric-Metal Capacitors with Atomic-Layer-Deposited Undoped HfO_2_ Films

**DOI:** 10.3390/ma15062097

**Published:** 2022-03-12

**Authors:** Chan-Hee Jang, Hyun-Seop Kim, Hyungtak Kim, Ho-Young Cha

**Affiliations:** 1School of Electrical and Electronic Engineering, Hongik University, 94 Wausan-ro, Mapo-gu, Seoul 04066, Korea; chanhee7208@g.hongik.ac.kr (C.-H.J.); hyunseop.kim@bristol.ac.uk (H.-S.K.); hkim@hongik.ac.kr (H.K.); 2Center for Device Thermography and Reliability (CDTR), University of Bristol, Bristol BS8 1TL, UK

**Keywords:** ferroelectric, undoped HfO_2_, metal-ferroelectric-metal, temperature, frequency, atomic layer deposition

## Abstract

In this study, we evaluated the temperature- and frequency-dependent ferroelectric characteristics of TiN/undoped HfO_2_/TiN metal-ferroelectric-metal (MFM) capacitors in which an undoped HfO_2_ film was deposited through atomic layer deposition (ALD). Successful ferroelectric characteristics were achieved after postdeposition annealing at 650 °C, which exhibited a remanent polarization of 8 μC/cm^2^ and a coercive electric field of 1.6 MV/cm at 25 °C (room temperature). The ferroelectric property was maintained at 200 °C and decreased as the temperature increased. The ferroelectric property was completely lost above 320 °C and fully recovered after cooling. The frequency dependency was evaluated by bias-dependent capacitance–voltage and s-parameter measurements, which indicated that the ferroelectric property was maintained up to several hundred MHz. This study reveals the ultimate limitations of the application of an undoped HfO_2_ MFM capacitor.

## 1. Introduction

Ferroelectric thin films have promising potential for various applications, including nonvolatile memories, energy-related devices, and negative capacitance field-effect transistors [1,2,3,4,5]. Although various ferroelectric materials, such as P(VDF-TrFE), Pb(Zr,Ti)O_3_ (PZT), and BaTiO_3_, have previously been intensively studied [2,5,6,7,8,9], HfO_2_ thin films have received significant attention recently owing to their excellent properties, such as high dielectric constant (20–25) and wide energy bandgap (~5.68 eV). Moreover, HfO_2_ thin films can be deposited using a complementary metal–oxide–semiconductor (CMOS) compatible atomic layer deposition (ALD) process [10,11,12,13,14]. Because ALD is based on self-limiting reactions leading to layer-by-layer growth, it exhibits a large-area uniformity and significant conformability, offering atomic-scale controllability [15]. The ferroelectric properties of HfO_2_ thin films have been reported with doping processes using various dopant elements, such as ZrO, Gd, Si, Al, Y, Sr, and La [4,5,16,17,18,19]. It has also been reported that the ferroelectric property can be achieved in undoped HfO_2_ films without the doping process [20,21,22,23]. Considering the deposition process, ALD doped HfO_2_ films require a higher deposition temperature than undoped HfO_2_ films to achieve the ferroelectric property [12,24]. Therefore, undoped HfO_2_ films have advantages of a lower temperature process and easy implementation without a delicate doping process [12,24,25].

Because of the mechanical dipole moment of ferroelectric materials, it is useful to determine the maximum temperature and frequency ranges to maintain the ferroelectric property of the MFM capacitor. Although various studies have been reported for different ferroelectric materials in a limited range of temperature and frequency [3,25,26,27,28,29,30,31,32], no study has been conducted with a MFM capacitor based on undoped HfO_2_. In this study, the ferroelectric property of an ALD-deposited undoped HfO_2_ thin film was successfully achieved via a postdeposition annealing process. Temperature-dependent polarization and frequency-dependent capacitance characteristics were measured up to >300 °C and GHz range, respectively, to determine the fundamental limitations of the ferroelectricity of TiN/undoped HfO_2_/TiN MFM capacitors.

## 2. Device Structure and Fabrication

TiN/undoped HfO_2_/TiN MFM capacitors were fabricated on a quartz substrate to eliminate the dielectric loss during high-frequency measurements. TiN electrodes play an important role in undoped HfO_2_ films. Unlike doped HfO_2_ films, the N impurity provided by metal nitrides is responsible for achieving the ferroelectric properties of undoped HfO_2_ films [33]. Figure 1a,b shows the top and cross-sectional schematics of the MFM capacitor, respectively, where a ground–signal–ground (GSG) pattern was employed for the s-parameter measurement.

The fabrication process of the device was as follows. After cleaning the substrates, a TiN bottom electrode with a thickness of 100 nm was deposited via RF sputtering. The films were deposited at a process pressure of 1 mTorr and an RF power of 600 W in Ar/N_2_ (10/25 sccm) ambient at two different temperatures, namely 25 °C (room temperature) and 250 °C. The sheet resistance of the TiN films deposited at room temperature and 250 °C were 9.89 and 9.33 Ω/sq, respectively. After patterning the bottom electrode, the TiN layer was etched using a fluorine-based plasma etching process. A 10 nm thick undoped HfO_2_ film was deposited via ALD at 220 °C with a TEMA-Hf precursor and O_3_ at a concentration of 100 gm^−3^. The deposition cycle consisted of a TEMA-Hf pulse time of 1 s, purge time of 20 s, O_3_ pulse time of 0.5 s, and purge time of 15 s. The growth per cycle was 1 Å/cycle, and the refractive index (RI) was 2.01. A TiN top electrode with a thickness of 100 nm was deposited on the undoped HfO_2_ film using RF sputtering. The top electrode was patterned and etched to complete the device structure. The diameter of the active region of the MFM capacitor was 100 μm. The fabricated MFM capacitors were annealed via rapid thermal annealing (RTA) at 650 °C for 1 min in a N_2_ atmosphere to achieve ferroelectric property. It was observed in our previous experiments that the optimum RTA temperature was 650 °C; the ferroelectric property was slightly weaker with RTA at 600 °C and noticeably reduced with RTA at 700 °C.

## 3. Results and Discussion

### 3.1. Ferroelectric Characteristics of TiN/Undoped HfO_2_/TiN MFM Capacitor

The polarization–electric field (P–E) characteristics of fabricated TiN/undoped HfO_2_/TiN MFM capacitors measured at 100 kHz before and after RTA are shown in Figure 2a,b, respectively. Although no hysteresis in the P–E characteristics was observed before RTA, successful ferroelectric property was achieved after RTA. No significant difference was observed in the P–E characteristics as a function of the deposition temperature for the TiN electrodes. When both the top and bottom electrodes were deposited at room temperature, the remanent polarization (P_r_) was 6.8 μC/cm^2^, and the coercive field (E_c_) was 1.46 MV/cm. When the electrodes were deposited at 250 °C, the values slightly decreased: P_r_ = 6.15 μC/cm^2^ and E_c_ = 1.2 MV/cm.

The permittivity–voltage characteristics of fabricated MFM capacitors were derived from the capacitance–voltage (C–V) characteristics measured at different frequencies. Figure 3 shows the frequency-dependent permittivity versus voltage characteristics for different samples; Figure 3d compares the maximum permittivity values as a function of the measurement frequency. The permittivity decreased as the frequency increased for all samples. The permittivity is determined by free dipoles oscillating in the presence of an alternating electric field. As the frequency increases, the dipoles begin to lag behind the electric field change, which decreases the permittivity [32]. As shown in Figure 3, the MFM capacitor with only the top electrode deposited at 250 °C exhibited relatively higher permittivity over the entire range of frequencies evaluated in this study. We speculate that the difference in permittivity was associated with unbalanced strains caused by the bottom and top TiN electrodes. Figure 4a,b shows the grazing-angle incident X-ray diffraction (GIXRD) patterns measured for two samples prepared with different TiN electrode deposition temperatures of room temperature and 250 °C, respectively. It was observed that a TiN (111) peak was dominant for the sample deposited at room temperature, whereas a TiN (200) peak was dominant for the sample deposited at 250 °C. Because of the different grain sizes and strains between TiN (111) and TiN (200) [34], the HfO_2_ film experienced unbalanced strains when the top TiN electrode was deposited at 250 °C, whereas the bottom TiN electrode was deposited at room temperature, which was responsible for the higher permittivity. It was reported that the strain increased the permittivity of the dielectric thin films [35,36]. It should be noted that new peaks appeared after the RTA process for both samples, which corresponded to the noncentrosymmetric o-phases that are responsible for the ferroelectric property [37].

The MFM capacitor with the bottom electrode deposited at room temperature and the top electrode deposited at 250 °C was chosen for the following study.

### 3.2. Temperature-Dependent Ferroelectric Property of TiN/Undoped HfO_2_/TiN MFM Capacitor

The temperature-dependent P–E characteristics of the MFM capacitor were evaluated in the temperature range from room temperature to 320 °C. As shown in Figure 4a, the ferroelectric hysteresis was maintained at 150 °C with a slight decrease in E_c_ and a slight increase in the saturation polarization (P_s_). The increased P_s_ is owing to the relaxation of oxygen vacancies, which can significantly contribute to their recombination, improving the ferroelectricity of the HfO_2_ film [28]. As shown in Figure 5a–c, significant deformation in the polarization characteristics occurred at 200 °C and significant degradation was observed at higher temperatures. The decreased P_r_ value at 200 °C is attributed to weaker spontaneous polarization caused by partial transition from the ferroelectric phase to the antiferroelectric phase and/or more defect formation at higher temperatures [28]. The hysteresis disappeared at temperatures higher than ~310 °C. Notably, this is the highest temperature at which an HfO_2_ MFM capacitor exhibits ferroelectric properties, although special composite materials, such as BaTiO_3_:Sm_2_O_3_, can achieve even higher operation temperature [3,28,29,30,31,38,39].

### 3.3. Frequency-Dependent Ferroelectric Characteristics of TiN/Undoped HfO_2_/TiN MFM Capacitor

To determine the frequency-dependent ferroelectric property of the fabricated MFM capacitor, C–V and s-parameter measurements were employed. The butterfly-shaped capacitance characteristics were measured at frequencies ranging from 10 kHz to 1 MHz, as shown in Figure 6a. The maximum capacitance value (C_MAX_) was obtained at a bias voltage of −0.5 V, whereas the minimum capacitance value (C_MIN_) was obtained at −3 V, and the capacitance tunability was defined by C_MAX_ − C_MIN_ [25]. Because conventional C–V measurements cannot be used at very high frequencies, the equivalent circuit of the MFM capacitor was extracted by s-parameter measurements; for high frequencies, it is convenient to describe a given network in terms of waves rather than voltages or currents [40]. The s-parameter measurements for the GSG pattern were performed in a frequency ranging from 100 MHz to 10 GHz using a network analyzer with the bias voltage conditions obtained for C_MAX_ and C_MIN_ at C–V measurements. The GSG-type MFM structure shown in Figure 1a can be modeled as capacitors connected in series, and the intrinsic ferroelectric capacitance can be extracted by converting the s-parameter into ABCD parameters [25,27]. Z_o_ in Equation (1) was 50 Ω in the s-parameter measurement.
(1)$$\mathrm{Capacitance}=\frac{1}{j\omega \times \mathrm{imag}\left(B\right)},\;\mathrm{where}\;B=Z_{o}\frac{\left(1+S_{11}\right)\left(1+S_{22}\right)-S_{12}S_{21}}{2S_{21}}$$

The C_MAX_ and C_MIN_ values extracted from the C–V and s-parameter measurements are plotted as functions of frequency in Figure 5b. Both the C_MAX_ and C_MIN_ values decreased as the frequency increased, reducing the capacitance tunability. Nevertheless, a capacitance tunability of up to several hundred MHz was achieved in this work. It is inferred that fabricated TiN/undoped HfO_2_/TiN MFM capacitor can be utilized as a variable capacitor up to hundreds of MHz, which can be used to explore a new field of undoped HfO_2_ MFM capacitors in microwave applications [41]. It is speculated that the rapid increase in capacitance at frequencies near 10 GHz is attributed to the series LC resonance.

## 4. Conclusions

In this study, the ferroelectric properties of TiN/undoped HfO_2_/TiN MFM capacitors were evaluated over a wide range of temperatures and frequencies. A 10 nm thick undoped HfO_2_ film was deposited via ALD, which was annealed at 650 °C after forming the electrodes. The fabricated MFM capacitor exhibited stable ferroelectric properties up to 150 °C with negligible degradation. Although the ferroelectric property weakened at temperatures higher than 200 °C, the hysteresis characteristics were maintained up to ~300 °C, which is the highest temperature reported for ferroelectric films. The frequency limitation was examined using C–V and s-parameter measurements, from which the capacitance tunability was achieved up to several hundred MHz. This study reveals the ultimate application conditions for a ferroelectric undoped HfO_2_ MFM capacitor.

## Figures and Tables

**Figure 1 materials-15-02097-f001:**
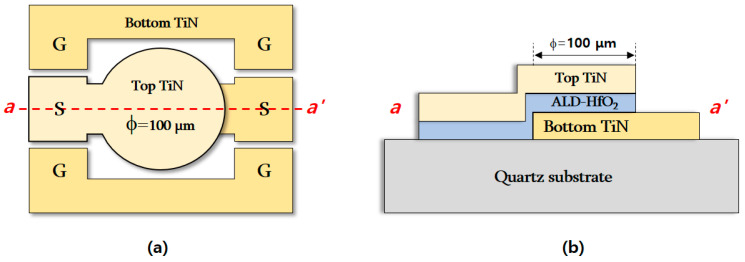
(**a**) Top view and (**b**) cross-sectional schematic along *a–a’* of TiN/undoped HfO_2_/TiN MFM capacitor fabricated on a quartz substrate.

**Figure 2 materials-15-02097-f002:**
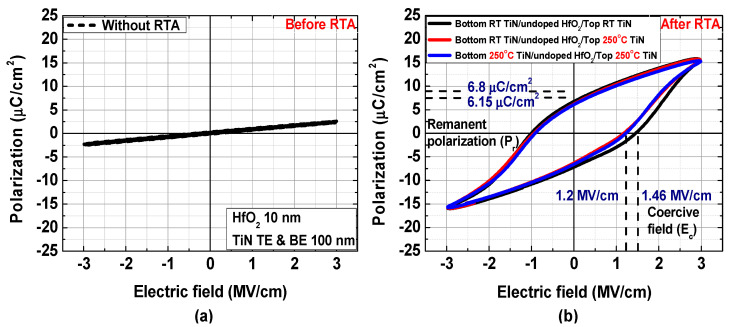
Polarization–electric field (P–E) characteristics of TiN/undoped HfO_2_/TiN MFM capacitors before (**a**) and after (**b**) RTA with different TiN deposition temperature conditions.

**Figure 3 materials-15-02097-f003:**
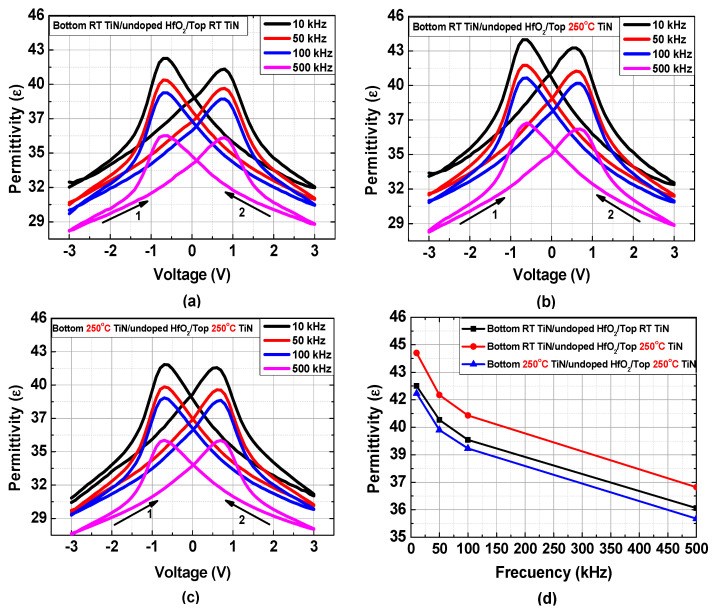
Frequency–dependent permittivity–voltage characteristics of MFM capacitors fabricated with different deposition temperatures for TiN electrodes. (**a**) Top and bottom electrodes deposited at room temperature, (**b**) bottom electrode at room temperature and top electrode at 250 °C, and (**c**) top and bottom electrodes at 250 °C. (**d**) Comparison of the maximum permittivity values for devices (**a**–**c**) as a function of measurement frequency.

**Figure 4 materials-15-02097-f004:**
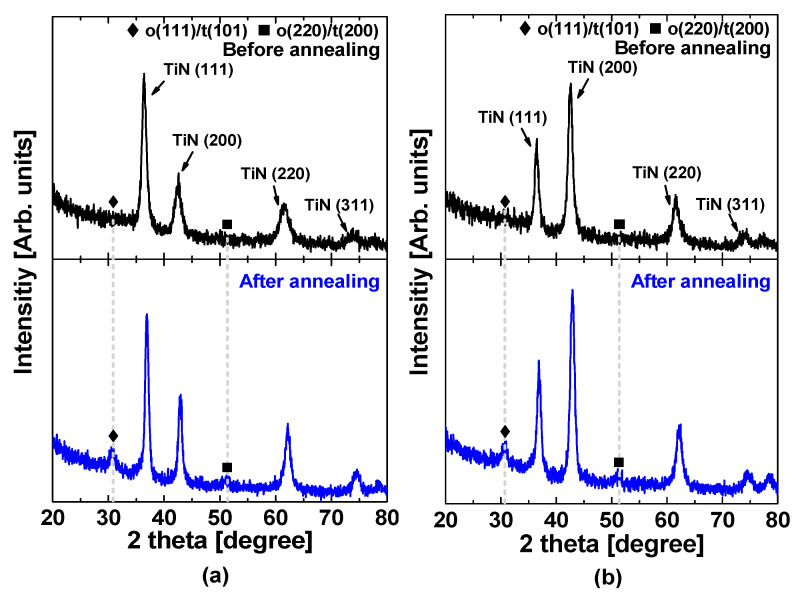
GIXRD patterns measured before and after the RTA process for TiN/undoped HfO_2_/TiN samples with both top and bottom TiN electrodes deposited at (**a**) room temperature and (**b**) 250 °C.

**Figure 5 materials-15-02097-f005:**
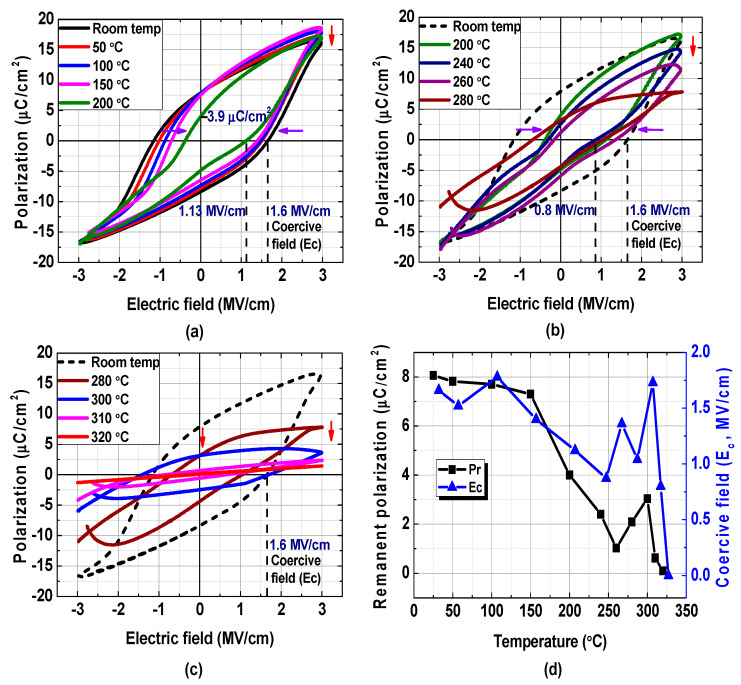
P–E characteristics of fabricated TiN/undoped HfO_2_/TiN MFM capacitor as a function of temperature: (**a**) from room temperature to 200 °C, (**b**) from 200 °C to 280 °C, and (**c**) from 280 °C to 320 °C. (**d**) P_r_ and E_c_ as functions of temperature.

**Figure 6 materials-15-02097-f006:**
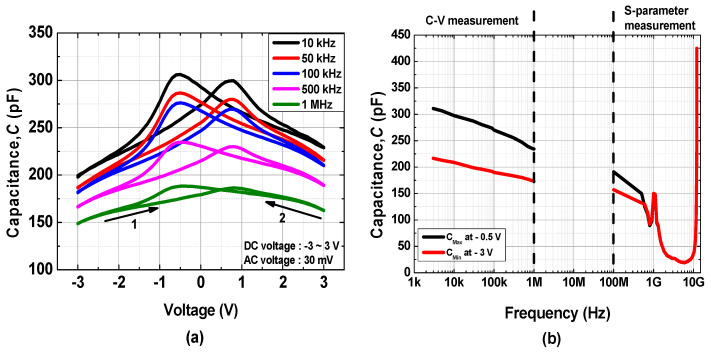
(**a**) Capacitance characteristics measured from 10 kHz to 1 MHz and (**b**) capacitance tunability characteristics versus frequency.

## Data Availability

Not applicable.

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
