# Peer review of "Temperature- and Frequency-Dependent Ferroelectric Characteristics of Metal-Ferroelectric-Metal Capacitors with Atomic-Layer-Deposited Undoped HfO2 Films"

_materials, 2022, doi:10.3390/ma15062097_

Round 1

Reviewer 1 Report

The authors investigated the temperature and frequency range limitations for the ferroelectricity of HfO2 film. Because the ferroelectric characteristics are based on dipole movement, it is obvious that such characteristics cannot be achieved at high temperature and/or high frequency. It is a bit surprising that such studies have not been reported in HfO2 films yet. I believe that this report will receive much attention from researchers in the field. However, I have the following questions that must be addressed clearly in the manuscript before publication.

  1. The authors mentioned about the tunability of capacitance at a very high frequency that the dipole movement would be insignificant. What could be the usage of the tunability for the device purpose? The authors should mention the potential usage of the tunability of ferroelectricity at the RF range. Are there any reports using the variable capacitors based on MFM? If yes, please include references.
  2. I assume that the characteristics in Fig. 2 were measured at low frequency. The authors must indicate the measurement frequency for Fig. 2.
  3. The authors claimed that this study exhibited the highest temperature range maintaining the ferroelectric characteristics without providing references. Please include references the authors compared. And, there must be other studies using different ferroelectric materials. Is HfO2 film better than others at the temperature limitation point of view?
  4. The MFM device was annealed at 650C to achieve the ferroelectric property. Is this the optimum temperature? How did the authors determine the annealing temperature.
  5. The text size of the legend in Fig. 2(b) is too small. Please enlarge the text size. And, the corceive field unit is placed below the number. Please rewrite them.
  6. In Fig. 3, the title of x-axis is mis-spelled. It must be not ‘volgate’ but ‘voltage’.
  7. What is the Zo value for Eq. (1)?
  8. In Fig. 5(a) and (b), the y-axis unit would be better to be consistent. Please use the unit of [pF] in the title in Fig. 5(a) and have numbers only in the values.

Author Response

We appreciate all reviewers’ valuable comments to improve our manuscript. With a limited time period, we did our best to provide additional discussion and data in the manuscript taking account of the reviewers’ comments carefully. Changes made in the manuscript except English correction are shown in blue in the revised manuscript. Please see our review responses below.

Reviewer 2 Report

This manuscript (materials-1612337)reports on the temperature and frequency-dependent ferroelectric characteristics of TiN/undoped-HfO2/TiN capacitors. There are some drawbacks, which should be corrected before publishing. Please see my detailed comments below.
1. The introduction did not emphasize the innovation of this work. Besides, the author should give some more interesting results except for the P-E property.
2. The author should specify the unit in right y-axis of Figure 4(d).
3. What is the actually meaning of “s-parameter measurements” in the abstract part. 
4. The author mentioned “the development of an undoped-HfO2 ferroelectric film is attractive in engineering because the appropriate dopant concentration ranges for achieving the ferroelectric property of doped-HfO2 films is small” in page 1, line 39, it is not clear and the author should give more demonstration for the exploration of un-doped films.
5. The author should add some description for the use of TiN electrode since more studies are focused on the common metal.
6. What is the mechanism for the permittivity-various characteristics with different deposition temperatures for TiN electrodes.
7. The English language must be improved. There are many mistakes in the language expression.

Author Response

(The authors gave the same response as above.)

Reviewer 3 Report

In this work, the authors investigated the temperature and frequency-dependent ferroelectric characteristics of TiN/undoped-HfO2/TiN metal-ferroelectric-metal (MFM) capacitors of which ferroelectricity persists up to 150 oC and hysteresis characteristics maintained up to ~300 oC. Atomic layer deposition was utilized to grow the undoped-HfO2 film with rapid thermal annealing (RTA). Top and bottom electrodes were deposited under room temperature and 250 oC to study their effects. High temperature and high frequency dependent ferroelectric properties were investigated to show the performance of MFM capacitor. However, this manuscript still needs to be improved. Therefore, I would not recommend its publication before addressing the following comments.

  1. Structural characterizations of TiN/undoped-HfO2/TiN metal-ferroelectric-metal (MFM) capacitors before and after RTA should be provided to tell structural M- and O-phases. For example, cross-sectional TEM images and X-ray diffraction patterns.
  2. RTA at 650 oC is the optimal for the ferroelectric properties of La-doped HfO2 (Appl. Phys. Lett. 108, 242905 (2016)). Does undoped HfO2 also share the same optimal RTA temperature that modifies oxygen vacancy concentration? The results of undoped HfO2 under other RTA temperatures should be provided.
  3. The top and bottom TiN electrodes were deposited at room temperature or 250 o Authors should discuss why the deposition temperatures of TiN electrodes affect the performance of MFM capacitors. What’s difference in structure between “Bottom RT TiN/un-doped HfO2/Top 250 oC TiN” and “Bottom 250 oC TiN/un-doped HfO2/Top 250 oC TiN” ?
  4. The reported Curie temperature for ferroelectric films as least can reach 330 oC in BaTiO3 (Nature Nanotechnology 6, 491–495 (2011)). So, the conclusion that the MFM shows highest operation temperature is not reasonable.

Author Response

(The authors gave the same response as above.)

Round 2

Reviewer 2 Report

 The presentation of this work is good, and it can be published in its current form.

Reviewer 3 Report

The authors addressed all comments well. I suggest to publish this manuscript.